

# Landscape and microhabitat features determine small mammal abundance in forest patches in agricultural landscapes

Luca Dorigo[1]   Francesco Boscutti[2]   Maurizia Sigura[2]

[1] Museo Friulano di Storia Naturale, Udine, Italy
[2] Di4A - Department of Agricultural, Food, Environmental and Animal Sciences, Università degli Studi di Udine, Udine, Italy

Corresponding authors
Luca Dorigo,
luca.dorigo1979@gmail.com
Maurizia Sigura,
maurizia.sigura@uniud.it

## ABSTRACT

Intensification of agricultural landscapes represent a major threat for biodiversity conservation also affecting several ecosystem services. The natural and semi-natural remnants, available in the agricultural matrix, represent important sites for small mammals and rodents, which are fundamental for sustaining various ecosystem functions and trophic chains. We studied the populations of two small mammals (*Apodemus agrarius, A. sylvaticus*) to evaluate the effects of landscape and habitat features on species abundance along a gradient of agricultural landscape intensification. The study was performed in Friuli Venezia Giulia (north-eastern Italy) during 19 months, in 19 wood remnants. Species abundance was determined using Capture-Mark-Recapture (CMR) techniques. In the same plots, main ecological parameters of the habitat (at microhabitat and patch scale) and landscape were considered. Abundance of *A. agrarius* increased in landscapes with high extent of permanent crops (i.e., orchards and poplar plantations) and low content of undecomposed litter in the wood understory. Instead, *A. sylvaticus*, a more generalist species, showed an opposite, albeit less strong, relationship with the same variables. Both species were not affected by any landscape structural feature (e.g., patch shape, isolation). Our findings showed that microhabitat features and landscape composition rather than wood and landscape structure affect populations' abundance and species interaction. The opposite response of the two study species was probably because of their specific ecological requirements. In this light, conservation management of agricultural landscapes should consider the ecological needs of species at both landscape and habitat levels, by rebalancing composition patterns in the context of ecological intensification, and promoting a sustainable forest patch management.

## INTRODUCTION

Land use change, intensification of agricultural practices, fragmentation of natural habitats and the consequent alteration of biological communities lead to a widespread decline in farmland biodiversity, measured across many different taxa (*Batáry et al., 2020*; *Benton,*

*Vickery & Wilson, 2003*; *Geiger et al., 2010*; *Kleijn et al., 2011*). In agricultural landscapes, a principal task for the conservation of farmland biodiversity and related ecosystem services is the understanding of the relationships between habitat features, habitat pattern and species population dynamics (*Fahrig, 2007*; *Le Roux et al., 2008*; *Tscharntke et al., 2005*). The need to preserve and enhance agricultural landscape diversity *via* management of non-crop habitats (*i.e.*, field margins, woodlots, hedgerows, and perennial grasslands) to support high levels of agricultural biodiversity is largely confirmed (*Carvell et al., 2011*; *Dainese et al., 2015*; *Dainese et al., 2016*; *De Simone, Sigura & Boscutti, 2017*).

Small forest patches may deliver several important ecosystem services to human society, but they receive little attention compared to large forest ecosystems. In particular, small forest patches represent central elements in the long-standing conflict between the need of agriculture productions and biological conservation (*Biber et al., 2015*; *Benayas, Bullock & Newton, 2008*).

In a recent review, *Decocq et al. (2016)* focused on the ecosystem services delivered by small forest patches of agricultural landscape highlighting their potential to support biodiversity both *in situ* (forested patch) and in the proximity area enhancing landscape connectivity. Here, patch structural features (pH of soil, nutrient availability, light) and habitat configurations patterns (*i.e.*, forest patch size, patch isolation) were defined as drivers for internal species richness and community composition.

Rodents and other small mammals are rarely target of conservation studies since they are often considered as pest or menace to production (*Brown et al., 2007*; *Butet, Paillat & Delettre, 2006*; Dickman 1999; *Putman, 1989*), thus associated with negative impacts on human activities (*Capizzi, Bertolino & Mortelliti, 2014*). However, small mammals are pivotal for important ecosystem functions (Dickman 1999). They contribute to soil aeration (*Laundré & Reynolds, 1993*) and play an important role as consumers of weeds and insects (*Bricker, Pearson & Maron, 2010*; *Gliwicz & Taylor, 2002*). Small mammals are crucial for diaspore dispersal of plants (Bogdziewicz et al. 2020; *Steele, Wauters & Larsen, 2005*) and fungi (*Capizzi & Santini, 2007*; *Sieg, 1987*). In several ecosystems they sustain food chains, representing the main prey biomass for several predator groups, such as birds, reptiles, and mammals (*Martin, 1994*; *Salamolard et al., 2000*; *Šálek et al., 2010*) and greatly contribute to the overall complexity of foodwebs (*Butet & Leroux, 2001*; *Korpimaki & Norrdahl, 1991*). For instance, low prey densities have been proved to affect raptor nesting densities in agricultural landscapes (see *e.g.*, *Michel, Burel & Butet, 2006*).

The populations of some small mammal species have experienced a decline, sometimes drastic, probably because of agriculture intensification (*Macdonald et al., 2007*; *Palmeirim, Santos-Filho & Peres, 2020*), acting at both landscape and patch levels. In particular, the intensification of agricultural landscapes has been shown to decrease the density of rare and stenoecious species in favor of generalist ones (*Burel et al., 2004*; *Gentili, Sigura & Bonesi, 2014*; *Millàn de la Peña et al., 2003*), provoking idiosyncratic responses. For example, *Silva, Hartling & Opps (2005)* and *Serafini, Priotto & Gomez (2019)* found that both structural complexity (*e.g.*, coverage and shape of forest patches) and landscape heterogeneity (*e.g.*, different land-use covers) promoted small mammal species diversity.

On the other hand, *Michel, Burel & Butet (2006)* demonstrated that, in areas where land-uses are relatively homogeneous (less landscape heterogeneity), species evenness but not species richness was negatively affected by the increasing of agricultural land-use intensification. At the patch scale, forest-floor characteristics, presence of coarse woody debris, understory vegetation, canopy composition (*Carey & Harrington, 2001*), forest age, vegetation complexity (*Pearce & Venier, 2005*), presence of deadwood (logs and rotten trunks) and litter structure (*Cox, Dickman & Cox, 2000*; *Kemper & Bell, 1985*; *Marsh & Harris, 2000*; *Simonetti, 1989*; *Szymański et al., 2020*; *Yahner, 1986*) can also influence small mammal occurrence. A high microhabitat complexity through high shrub diversity, high vegetation cover and low percentage of ground covered by bare-soil are known to increase overall small mammal abundance and species richness (*Gelling, Macdonald & Mathews, 2007*; *Silva & Prince, 2008*; *Szymański et al., 2020*).

Most of the previous studies evidenced strong effect of environmental drivers on small mammal. Nonetheless, comprehensive assessments at multiple ecological scales are still scarce and focus on species autecology. In this light, comparing species with different behavior (*i.e.,* generalist *vs* specialist) should be encouraged to get new insights into future agricultural landscape management.

In this study we investigated the effects of environmental features on small mammal abundance paying particular attention to forest specialists and habitat generalist species. We considered three different ecological scales: microhabitat, habitat (patch), and landscape. In particular, we tested the effects of landscape composition and configuration, forest structure and forest ground features on two small mammal species (*i.e., Apodemus sylvaticus*, and *A. agrarius*) in remnants of wood vegetation along an intensification gradient of agricultural land use. These species were chosen as they represent typical species of lowland woodlots, very common in the study area (*Dorigo, 2018*) and can easily be captured with live-capture systems.

Specifically, we ask two questions: (1) What are the relative and independent effects of heterogeneity of landscape elements, composition and configuration on small mammals abundance in woodlots within agricultural landscapes? (2) Does the effect of these mosaic properties differ between species with different ecological needs?

We expected to find a strong interplay between landscape composition and habitat features in determining small mammals species distribution and abundance. In particular, we expected to find contrasting responses between the species considered, hypothesizing that the generalist *A. sylvaticus* could be more affected by landscape composition and configuration rather than local forest characteristics.

## MATERIALS AND METHODS
### Study area, species and sampling design
The study was conducted in the agricultural landscape of Friuli Venezia Giulia region, North-East Italy (45°56′N, 12°47′E), an extensive lowland area (from 20 to 36 m a.s.l.) of alluvial deposits (*Fontana, 2006*). In the study area, a matrix of intensively-farmed arable fields of annual crops (winter cereals, *Zea mays,* and *Glycine max*) and permanent plantation (poplars and orchards) is discontinued by small hedgerows and woodlots.

Two small mammals species were considered. *Apodemus sylvaticus* (Linnaeus 1758) is a habitat generalist and opportunist species with high dispersal ability; it can occupy all woodlots and hedgerows, moving frequently between patches (*Bauchau & Le Boulengé, 1991*; *Fuentes-Montemayor et al., 2020*). Its main diet includes seeds, fruits, green parts of plants and fungi, but the animal component can be substantial only in some periods of the year (*Canova & Fasola, 1993*; *Hansson, 1985*). *Apodemus agrarius* (Pallas 1771) is a hygrophilous species which feeds on high-caloric food of animal and plant origins (*Babińska-Werka & Garbarczyk, 1981*). Its diet is quite varied and includes a wide range of invertebrates and plants (*Babinska-Werka, 1981*), in form of seeds and green parts (*Babinska-Werka, 1981*). In northern Italy, the species usually occurs at low altitudes, in the edges of mesophilous woods and in forest river banks (*Zulian, 1987*).

Sampling sites were 19 remnant woodlots not harvested in the past three years, characterized by medium-wet soils and smaller than 8 ha in size. The patches were selected along a gradient of landscape intensification ranging from 5% to 33% of semi-natural habitats with a 500 m buffer around each patch. The buffer size was established in agreement with the mobility of the most abundant small mammal species found (*Liro & Szacki, 1987*). The same distance was assumed to consider sampling patches as independent from each other. Patches with less than 500 m distance from each others were considered only if they are separated by barriers (*i.e.,* watercourses more than 5 m wide and one meter depth).

## Data collection
### Small mammal sampling
As trapping method we used Sherman traps (LFA type: cm 8 × 9 × 24. H.B. Sherman Traps©). This method of capture tends to be selective because it traps mostly small mammals weighing at least 5 grams like Murids (*i.e.*, *Apodemus*, *Mus,* and *Rattus* in the study area) (*Caceres, Nápoli & Hannibal, 2011*; *Umetsu, Naxara & Pardini, 2006*). It, therefore, precluded the capture of smaller insectivorous mammals. Within each sampling patch, 10 Sherman traps, spaced 10 m from each other, were placed along a 100 m line-transect. The traps were baited with pieces of fresh apple and sunflower seeds (*Tallmon & Mills, 1994*), to ensure the survival of trapped animals and were activated for four consecutive nights. Samplings during periods with excessive rainfall were excluded to avoid an excessive mortality of animals. To improve the control of temperature, each trap was lined on the bottom with dry plant material particularly important during the cold season.

In all areas (sites), four sampling sessions were repeated from April, 2013 to October 2014. Traps were checked once a day in the morning. Animals were trapped using the CMR (Catch-Mark-Recapture) method. Each individual was marked with an ear tag (Opivi Brand®), and its species, body mass, sex and age (juvenile, sub-adult and adult age classes) were recorded. The animals were released again at the same place where they were trapped. Trapping activities were authorized by the Italian Institute for Environmental Protection and Research (ISPRA) (Aut. PG/Ir Rif. Int. 26358-28967/2013) and by the Friuli Venezia Giulia region (Aut. Prot. SCPA/12.5/17552-2013).

### Microhabitat structure features

Microhabitat features were measured in sub-sample areas placed along the line transects with live traps (according to *Amori et al., 2015*). Around each trap, the ground cover was assessed within a buffer of 5 m. In particular, the cover percentage of herbaceous layer, rocky soil, bare ground, leaf litter and coarse woody debris (considering even small deposits of material also with diameter <3 cm) were visually estimated as percentage cover (*Amori et al., 2015*).

Finally, litter composition was evaluated, estimating the percentage of the categories: non-decomposed litter (whole leaves); partially decomposed litter (leaves partially degraded but leaf structure still recognizable); degraded litter (leaf structure not recognizable) in the litter stratum. This was carried out by two operators simultaneously around each trap, and the results were averaged.

### Wood structure features

At patch scale the presence of deadwood, understory stratification, tree diameter, woody and understory species and the presence of non-native tree and shrub species were assessed.

*Deadwood* Deadwood was measured in spring 2015 following mainly the protocol of *Bianchi et al. (2013)*. We measured with a caliper the diameters of all the logs (*e.g.*, stems, branches) (if >three cm). Length of the sample and decomposition state were also recoded. We used fixed area sampling (FAS) method (*Harmon & Sexton, 1996*), considered more reliable than other methods, especially with high amount of necromass composed of small fragments (*Bianchi et al., 2013*; Warren and Olsen 1964). Measures of deadwood were carried out in the central part of the forest along the transect used with traps for small mammals, around the traps 2, 5 and 8 of each transect.

Around selected traps a 5 m radius area (78.5 m$^2$) was surveyed. We opted for a lower threshold than indicated by *Bianchi et al. (2013)*, since the studied woodlots are of recent origin and deadwood is often composed of small size logs. Branches and woody fragments of diameters less than three cm were not considered in this protocol but were still evaluated as potential microhabitats (see Microhabitat, above). Each log (*e.g.*, stems, branches) of diameter >three cm was measured (length, major radius, minor radius) and referred to one of the three categories of the following scale: (i) standing deadwood, newly formed deadwood, with logs standing (*Marsh & Harris, 2000*); (ii) ground deadwood, deadwood on the ground; (iii) rotten deadwood, deadwood with spongy woody tissue and soaked with water.

The deadwood volume was calculated using the following Eq. (1):

$$V = \frac{\pi \cdot h}{3} \cdot \left[ R^2 + r^2 + (R \cdot r) \right] \tag{1}$$

where V is the volume, h is the height/length, R is the major radius, r is the minor radius (*Harmon & Sexton, 1996*). The total values of the deadwood, for each stage, were then averaged between the three FAS considered for each sampling station.

*Understory shrub density* We measured the renewal capacity and the vertical stratification of the understory layer of the studied woods. In particular, we identified, along the transect

used with the traps for the small mammals samplings (around the trap 2, 5 and 8 of each transect), circular areas of 2 m radii (6.28 m$^2$). Within each area, all woody plants with diameter of less than 7.5 cm have been recorded, indicating the species, and measuring the stem diameter at ground level.

*Trees density and diameter* In the center of the transect (trap 5), we placed a 10x10 m plot, inside of which we noted the number of woody plants with more than 7.5 cm in diameter. For each plant we reported the species and the diameter at breast height (D.B.H). All data were finally pooled as the average diameter of each plot, considering their standard deviation as a proxy of plot variability.

*Presence of alien species* We verified the presence of non-native arboreal plants within the $10 \times 10$ m plots (see above: *Trees density and diameter*) as well as the presence and the number of non-native shrubby plants inside the 2 m radii plots (see above: *Understory shrub density*).

### Landscape structure features

Land use was mapped in two concentric buffers, respectively of 500 m (0.785 km$^2$) and 250 m (0.196 km$^2$) radii, around the 19 surveyed remnant woodlots. Land use types were mapped using aerial images of the study area (year 2014) in a Geographic Information System (GIS) open source environment (QGIS). Mapped land use classes were (1) arable lands, *i.e.,* annual crops; (2) permanent crops, *i.e.,* orchards, vineyards and poplars; (3) set aside, *i.e.,* fallow lands, uncultivated areas with herbs and shrub cover; (4) woods, *i.e.,* hedgerows and woodlots; (5) settlements, *i.e.,* urban areas, industrial areas and isolated buildings. Rivers and roads were not considered as polygons but measured as linear elements (length).

Landscape pattern was analyzed by means of landscape metrics considering features at both patch and landscape scales (composition and configuration). For this purpose, selected variables (*McGarigal & Marks, 1995*): woodlot surface area; woodlot perimeter; woodlot Perimeter-Area Ratio (k =p/a); Patch Shape Index ($PSI = \frac{Perimeter}{\sqrt{2 \cdot \pi \cdot Area}}$); Mean proximity index for all patches in the landscape (MPI); mean proximity index for patches comprising the wood class (MPI_Wood); Mean nearest-neighbor distance for patches comprising the wood class (MNN); Interspersion and juxtaposition index for patches comprising the wood class (IJI); Total number of wood patches in the landscape (NumP); Mean patch size for patches comprising the wood class (MPS); Total edge for patches comprising the wood class (TE); Percent of landscape occupied by each patch type in the landscape (PLAND); Modified Simpson's diversity index based on *Pielou (1975)* modification of Simpson's diversity index (MSIDI). Landscape structure analyses were performed using the Fragstat software (*McGarigal, Cushman & Ene, 2012*).

## Data analyses

Low re-capture probability, and thus unreliable estimates of abundance, did not allowed us to use estimates of abundance taken from capture-recapture models, therefore an abundance index was used. Population abundance was calculated by the relative abundance

index (RAI), used as a proxy measure of population density (*Cagnin et al., 1998*; *Klaa, Mill & Incoll, 2005*; *Ouin et al., 2000*) according to the formula Eq. (2).

$$RAI = \frac{Ns}{ns} \cdot ts \cdot 100 \tag{2}$$

where $s$ is the sampling site, $N$ is the number of individuals of each species, $n$ is the number of nights that a trap is open for, and $t$ is the number of active traps.

For each site, 40 environmental variables were measured, grouped in microhabitat structure (8), wood structure (13), landscape composition (6) and landscape configuration (13). All analyses were performed using R statistical software (*R Core Team, 2021*).

All the variables were initially analyzed and selected to cope (multi-) collinearity and to obtain more parsimonious models (*Faraway, 2005*). Variable collinearity was assessed through Principal Component Analysis (PCA) and Pearson correlation (Pearson's $r < 0.6$), using the 'car' package (*Fox & Weisberg, 2011*). After this selection, 19 variables were kept as environmental predictors, divided in four groups: (i) microhabitat structure, *i.e.,* litter cover percentage, coarse woody debris percentage, bare ground percentage, and undecomposed litter percentage; (ii) wood structure, *i.e.,* tree species number, tree DBH, tree density, shrub density, number of shrub species, total deadwood, and rotten deadwood; (iii) landscape composition, *i.e.,* percentage of arable land, woods, permanent crops, settlements, and set aside; (iv) landscape configuration, *i.e.,* area/perimeter ratio, PSI, and MPI_Wood.

Multi-model inference (MMI) within an information theoretic framework was used to evaluate the influence of environmental variables on the abundance of *Apodemus agrarius* and *A. sylvaticus*, at the different scale (*Burnham & Anderson, 2002*). This technique compares the fit of all possible models (including the null model) obtained by the combination of the variables with the "dredge" function in the MuMIn package (Bartoń, 2015) of the R software. We used Akaike's information criterion (AIC) (*Akaike, 1973*) to choose the best fitting model. AIC measures the relative quality of a model dealing with the trade-off between the complexity of the model and the goodness of fit. The best fit is indicated by the lowest AIC value (AIC MIN). In a set of models each model ($i$) can be ranked using its difference in AIC score to the best-fitting model ($\Delta$ AIC$i$ = AIC$i$- AIC$i$ MIN). A model in the set can be considered plausible if its $\Delta$AIC is below 2 (*Burnham & Anderson, 2002*). We also derived the Akaike's model weight ($w_i$) which is the probability that the model ($i$) is the best-fitting model if data were collected again under identical circumstances (*Burnham & Anderson, 2002*). We also calculated the relative importance of the variables using Akaike's model weight.

Linear mixed-effects models (LMMs) were used to estimate model parameters as model residuals approximated a normal distribution. Models included population abundance as response variable and species, environmental variables, and their interactions as fixed effects. The transect ID was included as random effect. Because of the low number of real replicates, the four groups of variables linked with the three scale levels (microhabitat, habitat, landscape composition + configuration) were modeled separately. An overall model was performed using only the variables which had a significant averaged coefficient ($p < 0.05$) and a high Relative Variable Importance (RVI) value (RVI>0.7). LMMs were
conducted for each response variable separately using both a linear model with and without including a quadratic term to account for a possible non-linear relationship. The multi-model inference based on AIC was executed with the 'MuMIn' package (*Bartoń, 2016*). The LMMs were applied using the "nlme" package (*Pinheiro et al., 2017*). Model goodness-of-fit was further tested using diagnostic plots of residuals and goodness indices (*Faraway, 2005*; *Zuur et al., 2009*).

# RESULTS

In all sampling sessions 868 captures were recorded, of which 36.5% were recaptured individuals. The total capture effort was 3024 trap-nights, with a capture index (Ic = N ×100 catch / trap-nights) of 28.7. Among the catches, 55.3% was represented by *A. agrarius*, 40.1% *of A. sylvaticus*, 4.1% by *Rattus norvegicus* and the remaining 0.5% by *R. rattus, Microtus arvalis* and *Crocidura suaveolens*.

*Apodemus sylvaticus* was caught in all sampling sites while *A. agrarius* occured in 16 sites and, together, these species represented about 95% of the individuals caught. Both rodent species were caught in all sampling sessions. For all the investigated scales the null models showed a low weight (weight <0.001) and were not considerable as plausible (delta AIC >12).

## Microhabitat structure

For microhabitat structure, the multi-model inference showed that eight models were supported (Table 1). The models included the percentage of deadwood, ground cover of bare soil, percentage of undecomposed litter, species, and their interaction. The most important variable was the percentage of undecomposed litter and its interaction with species (RVI>0.70; $p = 0.05$). Albeit not significant, presence of deadwood and ground cover of bare soil were also important (RVI =0.71 and 0.68, respectively), whereas their interactions with species were negligible. In general, small mammals abundance decreased as undecomposed litter increased. Models explained between 27 and 29% of the total variation in small mammal abundance.

## Wood structure

For the group of variables describing the wood structure, the multi-model inference showed that there were seven plausible models, which included tree density, DBH, tree species richness, shrub richness, small mammal species and their interactions (Table 2). Among these variables, tree density, number of tree species (and its interaction), number of shrub species (and its interaction) showed a high relative importance (RVI>0.7), albeit none of them was significant. The models explained between 20 and 28% of the total variation in small mammal abundance.

## Landscape configuration

The analysis was conducted for both 500 m and 250 m.

Using landscape configuration variables in 250 m buffers, three models were supported (Table 3), which included the following variables: area/perimeter ratio, PSI, species. The

**Table 1  List of plausible models performed with multi-model inference for microhabitat structure.** The estimates are reported: the intercept (Int), the variables considered in each model (ddW = mean percentage of surface around the traps occupied by deadwood; grn = mean percentage of surface around the traps occupied by bare soil; spcs = species; und = mean percentage of undecomposed litter in the litter stratum), $R^2$, AIC, $\Delta AIC$ and model weight (wi). Relative Variable Importances values and significance are shown in bold.

| (Int) | ddW | grn | spcs | und | ddW:spcs | grn:spcs | spcs:und | $R^2$ | AIC | delta | weight |
|---|---|---|---|---|---|---|---|---|---|---|---|
| 126.7 | −0.25 | 0.03 | + | −1.30 | + | + | + | 0.29 | 344.6 | 0 | 0.11 |
| 127.5 | −0.26 | | + | −1.31 | + | | + | 0.29 | 344.9 | 0.31 | 0.10 |
| 127.4 | −0.26 | 0.00 | + | −1.31 | + | | + | 0.29 | 345.4 | 0.87 | 0.07 |
| 120 | | 0.13 | + | −1.25 | | + | + | 0.28 | 345.6 | 1.04 | 0.07 |
| 122.4 | | | + | −1.27 | | | + | 0.27 | 345.6 | 1.05 | 0.07 |
| 111.3 | 0.33 | 0.26 | + | −1.17 | | + | + | 0.28 | 345.9 | 1.38 | 0.06 |
| 116.1 | 0.33 | | + | −1.23 | | | + | 0.28 | 346.2 | 1.59 | 0.05 |
| 124.8 | | −0.13 | + | −1.30 | | | + | 0.27 | 346.4 | 1.8 | 0.05 |
| | **0.71** | **0.68** | **1** | **0.86 .** | **0.46** | **0.44** | **0.72 .** | | | | |

**Table 2  List of plausible models performed with multi-model inference at wood structure scale.** The estimates are reported: the intercept (Int), the variables considered in each model (Tree_dns = tree density; Mean_dmt = mean trunk diameter; Tree_spcs = number of tree species; Shr_spcs = number of shrub species; spcs = species), $R^2$, AIC, $\Delta AIC$ model weigh AIC and model weight (wi). Relative Variable Importances values and significance are shown in bold.

| (Int) | Tree_dns | Mean_dmt | Tree_spcs | Shr_spcs | spcs | Tree_dns: spcs | Mean_dmt: spcs | Tree_spcs: spcs | Shr_spcs: spcs | $R^2$ | AIC | delta | weight |
|---|---|---|---|---|---|---|---|---|---|---|---|---|---|
| 95.33 | 0.57 | −1.01 | −6.66 | −1.89 | + | + | + | + | + | 0.28 | 333.7 | 0 | 0.23 |
| 75.24 | 0.94 | | −6.98 | −2.90 | + | + | | + | + | 0.22 | 334.1 | 0.36 | 0.19 |
| 94.61 | 0.60 | −1.01 | −6.66 | −1.85 | + | | + | + | + | 0.28 | 334.8 | 1.1 | 0.13 |
| 84.17 | 0.77 | −0.45 | −6.8 | −2.45 | + | + | | + | + | 0.24 | 334.8 | 1.11 | 0.13 |
| 106.4 | | −1.06 | −6.74 | −2.56 | + | | + | + | + | 0.26 | 334.9 | 1.21 | 0.12 |
| 78.33 | 0.77 | | −7.01 | −3.13 | + | | | + | + | 0.22 | 335.2 | 1.46 | 0.11 |
| 92.39 | | | −7.15 | −4.14 | + | | | + | + | 0.20 | 335.6 | 1.83 | 0.10 |
| | **0.75** | **0.66** | **0.99** | **0.95** | **1** | **0.48** | **0.42** | **0.9** | **0.83** | | | | |

**Table 3  List of plausible models performed with multi-model inference for landscape configuration in 250 m buffer.** The estimates are reported: the intercept (Int), the variables considered in each model (Are_prm = area/perimeter ratio of wood patches, PSI = patch shape index), $R^2$, AIC, $\Delta AIC$ and model weight (wi). Relative Variable Importances values and significance are shown in bold.

| (Int) | Are_prm | PSI | spcs | Are_prm:spcs | PSI:spcs | $R^2$ | AIC | delta | weight |
|---|---|---|---|---|---|---|---|---|---|
| 38,7 | | 3,52 | + | | + | 0,06 | 342,9 | 0 | 0,45 |
| 26,01 | 0,43 | 5,17 | + | | + | 0,1 | 343,7 | 0,83 | 0,3 |
| 32,87 | 0,2 | 4,28 | + | | + | 0,11 | 344,1 | 1,19 | 0,25 |
| | **0,55** | **0,99** | **1** | **0,25** | **0,91** | | | | |

most important, albeit not statistically significant, was Patch Shape Index and its interaction with species (RVI>0.91). Models explained between 6 and 11% of the total variation in small mammals abundance.

Considering the same descriptors within 500 m buffers, seven plausible models were obtained, including the variables: area/perimeter ratio, PSI, MSI, species and their interactions with the variables (Table 4). Also in this case, no variable showed a high

**Table 4 List of plausible models performed with multi-model inference for landscape configuration in 500 m buffer.** The estimates are reported: the intercept (Int), the variables considered in each model (Are_prm = area/perimeter ratio of wood patches, PSI = patch shape index, MSI_500 = Modified Simpson's Diversity Index in 500 m buffer), $R^2$, AIC, $\Delta AIC$ and model weight (wi). Relative Variable Importances values and significance are shown in bold.

| (Int) | Are_prm | MSI_500 | PSI | spcs | Are_prm:spcs | MSI_500:spcs | PSI:spcs | $R^2$ | AIC | delta | weight |
|---|---|---|---|---|---|---|---|---|---|---|---|
| 38,7 | | | 3,52 | + | | | + | 0,06 | 342,9 | 0 | 0,19 |
| −18,15 | | 0,57 | 6,78 | + | | + | + | 0,2 | 343,3 | 0,41 | 0,15 |
| 26,01 | 0,43 | | 5,17 | + | | | + | 0,09 | 343,7 | 0,83 | 0,12 |
| −15,68 | −0,15 | 0,59 | 6,31 | + | + | + | + | 0,26 | 343,9 | 1,04 | 0,11 |
| 32,87 | 0,2 | | 4,28 | + | + | | + | 0,11 | 344,1 | 1,19 | 0,1 |
| −23,34 | 0,31 | 0,53 | 7,75 | + | | + | + | 0,22 | 344,7 | 1,84 | 0,07 |
| 15,32 | | 0,23 | 4,86 | + | | | + | 0,11 | 344,8 | 1,88 | 0,07 |
| | **0,54** | **0,55** | **0,99** | **1** | **0,27** | **0,39** | **0,91** | | | | |

**Table 5 List of plausible models performed with multi-model inference for landscape composition in 250 m buffer.** he estimates are reported: the intercept (Int), the variables considered in each model (crp_250 = percentage of surface area occupied by crops; prm_250 = percentage of surface area occupied by permanent crops; stt_250 = percentage of surface area occupied by settlements), $R^2$, AIC, $\Delta AIC$ and model weight (wi). Relative Variable Importances values and significance are shown in bold.

| (Int) | crp_250 | prm_250 | stt_250 | spcs | crp_250:spcs | prm_250:spcs | stt_250:spcs | $R^2$ | AIC | delta | weight |
|---|---|---|---|---|---|---|---|---|---|---|---|
| 32.16 | | 0.96 | −0.46 | + | | + | | 0.36 | 345.3 | 0 | 0.29 |
| 27.81 | | 1.01 | | + | | + | | 0.30 | 345.3 | 0.06 | 0.28 |
| 34.72 | | 0.92 | −0.73 | + | | + | + | 0.37 | 345.6 | 0.38 | 0.24 |
| 90.76 | −0.77 | | −1.4 | + | + | | + | 0.38 | 346 | 0.75 | 0.20 |
| | **0.44** | **0.8 \*** | **0.72** | **1** | **0.28** | **0.71 \*** | **0.43** | | | | |

RVI value (>0.7), with the exception of the PSI (and its interaction with the species). The most important, albeit not statistically significant, was PSI and its interaction with species (RVI>0.91). Models explained between 6 and 26% of the total variation in small mammals abundance.

## Landscape composition

The analysis was conducted for both 500 m and 250 m radius buffers.

Using landscape composition variables in 250 m buffers, four models were supported (Table 5). These models included crop percentage, permanent crop percentage, and settlement percentage. The most important variable was the percentage of permanent crops and its interaction with species (RVI>0.70; $p = 0.05$). Settlement percentage was also important among models (RVI =0.72), but not statistically significant.

Considering landscape composition variables within 500 m buffers, three models were supported (Table 6). These models included crop percentage, permanent crop percentage, and settlement percentage. The most important variable was the percentage of permanent crops and its interaction with species (RVI>0.89; $p = 0.05$).

## Overall model

The multi-model inference applied to the selected variables from each environment structural level (overall model) showed one supported model both in 250 m and 500 m

**Table 6   List of plausible models performed with multi-model inference for landscape composition in 500 m buffer.** The estimates are reported: the intercept (Int), the variables considered in each model (crp_500 = percentage of surface area occupied by crops; prm_500 percentage of surface area occupied by permanent crops; stt_500 = percentage of surface area occupied by settlements), $R^2$, AIC, $\Delta AIC$ and model weight (wi). Relative Variable Importances values and significance are shown in bold.

| (Int) | crp_500 | prm_500 | stt_500 | spcs | crp_500:spcs | prm_500:spcs | stt_500:spc | $R^2$ | AIC | delta | weight |
|---|---|---|---|---|---|---|---|---|---|---|---|
| 24,08 | | 1,25 | | + | | + | | 0,31 | 343,9 | 0 | 0,25 |
| 29,18 | | 1,14 | -0,38 | + | | + | | 0,35 | 344,8 | 0,95 | 0,15 |
| 0,96 | 0,34 | 1,49 | | + | | + | | 0,34 | 345,1 | 1,26 | 0,13 |
| | **0,47** | **0,94\*** | **0,51** | **1** | **0,21** | **0,89\*** | **0,21** | | | | |

**Table 7   List of plausible models performed with multi-model inference in 250 m buffer.** The estimates are reported: the intercept (Int), the variables considered in each model (prm_250 = percentage of surface area occupied by permanent crops; spcs = species; und = undecomposed litter percentage), $R^2$, AIC, $\Delta AIC$ and model weight (wi). Relative Variable Importances values and significance are shown in bold.

| (Int) | prm_250 | spcs | und | prm_250:spcs | spcs:und | $R^2$ | AIC | delta | weight |
|---|---|---|---|---|---|---|---|---|---|
| 20,68 | 0,1873 | + | -0,267 | + | + | 0,56 | 214,8 | 0 | 0,748 |
| | **0,73\*\*\*** | **0,91.** | **0,81\*\*\*** | **0,7\*\*\*** | **0,75\*\*\*** | | | | |

**Table 8   List of plausible models performed with multi-model inference in 500 m buffer.** The estimates are reported: the intercept (Int), the variables considered in each model (prm_500 = percentage of surface area occupied by permanent crops; spcs = species; und = undecomposed litter percentage), $R^2$, AIC, $\Delta AIC$ and model weight (wi). Relative Variable Importances values and significance are shown in bold.

| (Int) | prm_500 | spcs | und | prm_500:spcs | spcs:und | $R^2$ | AIC | delta | weight |
|---|---|---|---|---|---|---|---|---|---|
| 101,5 | 1,267 | + | -1,295 | + | + | 0,51 | 336,9 | 0 | 0,904 |
| | **0,99\*\*\*** | **1** | **0,97\*\*** | **0,98\*\*** | **0,91\*** | | | | |

buffers (Tables 7, 8). The models confirmed the importance of the extension of permanent crops and the undecomposed litter layer, highlighting a significant interaction of the study species with the variables ($p < 0.001$; RVI>0.7). In particular, where the landscape had a higher percentage of permanent crops, the abundance of *A. agrarius* increased, whereas the abundance of *A. sylvaticus* decreased (Fig. 1A). Instead, an increase of undecomposed litter favors *A. sylvaticus*, decreasing the abundance of *A. agrarius* (Fig. 1B). The models including variables within the 250 and 500 m buffers explained respectively 56 and 51% of the total variation in small mammals abundance.

## DISCUSSION

Our findings shed new light on the interplay between different ecological scales in determining small mammals abundance and distribution. We found species with different ecological niches to differ significantly in landscape and habitat features, suggesting the occurrence of alternative strategies for species coexisting in fragmented forest patches in agricultural landscapes.

Considering the contribution of each variable group and their selection, we found both landscape composition and microhabitat features to significantly affect species abundance, with contrasting species responses. On the contrary, wood structure did not directly influence the abundance of the two studied species, not confirming previous studies

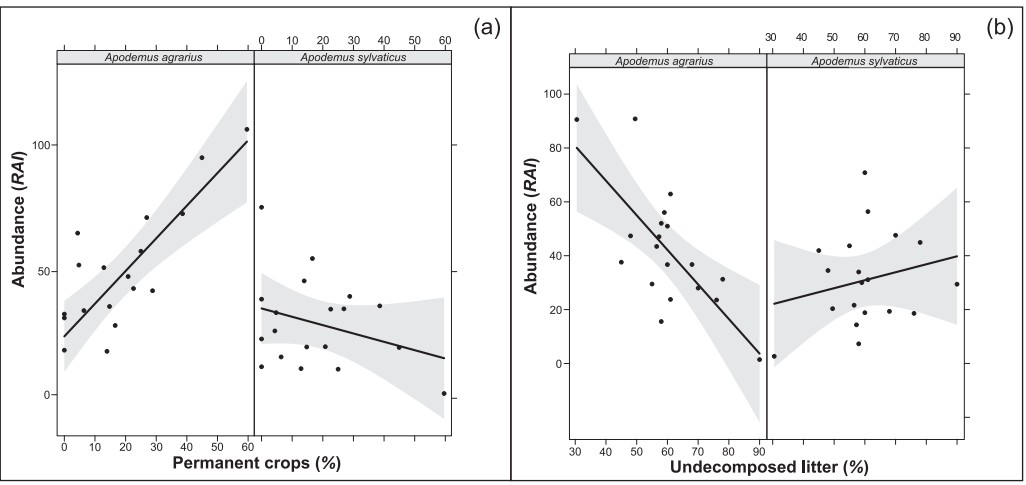

**Figure 1 Relationships between population abundance, species and their interactions with environmental variables (A: permanent crops; B: undecomposed litter).** Relationships between population abundance, species and their interactions with environmental variables, calculated with the best fitting model chosen within Multi-model inference. (A) Permanent crops. (B) Undecomposed litter. Permanent crops = proportion of area covered by permanent crops (poplars, orchards) in buffer area of 250 m around the sampling plot; undecomposed litter = mean percentage of undecomposed litter in the litter stratum. Confidence intervals (95%) are also shown.

evidencing significant relationships between abundance of small mammal forest species and forest tree density (*Capizzi & Santini, 2007*) and size (*Capizzi, Battistini & Amori, 2003*). Albeit not significant, our results also highlighted positive relationships between small mammal abundance and tree density, as well as number of tree species and number of shrub species, that could be interpreted as direct relationship with the complexity of the ecological niche. Several studies reported a strong relationship between deadwood abundance and the presence of small mammals. In fact, deadwood represents a source of potential shelters and hosts large invertebrate communities which are potential sources of food (*Amori et al., 2015*; *Bellocq & Smith, 1997*; *Bowman et al., 2000*; *Harmon et al., 1986*; *Kemper & Bell, 1985*; *McCay, 2000*; *Miller & Getz, 1977*; *Szymañski et al., 2020*; *Tallmon & Mills, 1994*; *Yahner, 1986*). Several studies also indicate that the decay stage may be important in habitat selection of small mammals (*Bowman et al., 2000*; *McCay, 2000*). Availability of understory cover is also a key condition for small mammals (*Barry, Botje & Grantham, 1984*; *Bellocq & Smith, 1997*; *Carey & Harrington, 2001*; *Cox, Dickman & Cox, 2000*; *Marsh & Harris, 2000*; *Simonetti, 1989*). Probably its importance is connected with food availability and the related possibility to reduce either the areas of predation (*Bellocq & Smith, 1997*; *Miller & Getz, 1977*; *Simonetti, 1989*) or the competition, by expanding the available space (*Montgomery, 1980*). Regarding microhabitat structure, the bare soil cover, presence of deadwood and litter structure are known to affect small mammal populations (*Amori et al., 2015*; *Cox, Dickman & Cox, 2000*; *Denny et al., 2021*; *Kemper & Bell, 1985*; *Marsh & Harris, 2000*; *Simonetti, 1989*; *Yahner, 1986*). Our results confirmed these findings, but highlighted the pivotal role of the litter decomposition stage. Litter

decomposition rate is driven by environmental conditions, chemical composition of soil and intensity of soil organism activity (*Aerts, 1997*; *Harmon et al., 1986*).

The speed and degree of degradation of leaves depends mainly on the species of tree (*He et al., 2019*; *Mathews & Kowalczewski, 1969*; *Nykvist, 1962*) and other environmental variables such as temperature and humidity. In addition, the presence of large invertebrates such as earthworm, millipedes and other taxa is also relevant, enhancing the degradation of organic matter (*Bocock, 1964*; *Mathews & Kowalczewski, 1969*; *Tresch et al., 2019*). The dynamics of the soil degradation process is connected to invertebrate biodiversity: well developed temperate woodland soils can host up to a hundred species of animals (*Anderson, 1975*; *Tresch et al., 2019*). Moreover, leaf litter quality affects numerous trophic levels such as insects, other invertebrates (*Koivula et al., 1999*; *Sabo, Soykan & Keller, 2005*) and small mammals (*Canova & Fasola, 1991*; *Hansson, 1978*; *Kaminski et al., 2007*). Litter cover affects the use of habitat by black rats (*Cox, Dickman & Cox, 2000*), which spend most of the time foraging on the ground, probably obtaining food from the litter (*Cox, Dickman & Cox, 2000*).

A dense litter cover would produce suitable mesic conditions in soil and litter ensuring appropriate microenvironments for litter-dwelling arthropods used as food (*Yahner, 1986*). Wood mice catch and find invertebrate prey, which are an important source of food, under litter cover (*Piper, Lewis & Compton, 2014*). The microclimate conditions of the litter can influence movements of small mammals, whom sounds are more difficult to hear in moist than in dry litter, influencing predation (*Vickery & Bider, 1981*). Predation risk is considered one of the most important factors determining selection of microhabitat in small mammals (*Bellocq & Smith, 1997*; *Fragoso, Santos-Reis & Rosalino, 2020*; *Simonetti, 1989*).

Despite this evidence, we found a different interaction of the two considered species with the environment conditions. In fact, abundance of *A. agrarius* was inversely related with the percentage of undecomposed litter, while the opposite trend was observed in *A. sylvaticus.*

Probably *A. agrarius* is more dependent on litter quality because it generally avoids xeric conditions, as reported by *Yahner (1986)* for other species in relation to the moisture of the habitat. This species is known to prefer habitats with dense soil cover and it usually moves through the litter stratum because of its fossorial behaviour (*Kuncová & Frynta, 2009*), probably to escape predators (*Orlandi & Paolucci, 2004*).

On the contrary, *A. sylvaticus* occurs in both woody and in open habitats (*Heroldová et al., 2007*) avoiding cut crops or cut set aside (*Tattersall et al., 2001*) and bare soil (*Tew, Todd & Macdonald, 2000*), showing this species is poorly influenced by the structure of litter, in disagreement with what is reported in other studies (*e.g.*, *Balestrieri et al., 2017*).

Moreover, *A. sylvaticus* is apparently not affected by wood characteristics probably because this ubiquitous species prefers the diversity of the complex mosaic created by agricultural landscape rather than the continuous forest habitat (*Geuse, Bauchau & Boulengé, 1985*; *Kozakiewicz et al., 1999*).

It is well known that composition of landscape can influence the population dynamics of small mammals (*Fischer & Schröder, 2014*; *Silva, Hartling & Opps, 2005*). Our results
confirmed this evidence pointing out the positive role of some non-natural habitats (*i.e.*, permanent crops) in agreement with what was found in other studies (*Fitzgibbon, 1997*; *Macdonald et al., 2007*; *Tattersall et al., 1999*; *Vieira et al., 2009*). Even though some studies showed small mammal abundances to be related with landscape structure (*Fischer, Thies & Tscharntke, 2011*; *Kozakiewicz et al., 1999*; *Macdonald et al., 2007*; *Van Apeldoorn et al., 1992*; *Vieira et al., 2009*), in our study, no significant relationship emerged considering landscape configuration. We observed a positive relationship between the abundance of *A. agrarius* and the presence of permanent crops, despite the percentage of natural woods was relatively low in the studied area (ca. 8% on average). This result suggests that permanent woody crops, such as poplars and orchards, may be important habitats for the study species, especially if placed in the surroundings of woodlots, offering canopy cover and in some cases a dense leaf litter. On the contrary, we did not observe such strong relation for *A. sylvaticus*, which seems less influenced by the presence of permanent crops in the landscape but with a negative relationship. It seems that *A. sylvaticus* in Mediterranean areas uses mostly nests positioned far away from orchards, probably to avoid predation (*Rosalino et al., 2011*), although other studies show that the species is quite common in poplar plantations (*Balestrieri et al., 2017*) and orchards (*Dickman & Doncaster, 1987*).

The differences found for the two species could be linked to their ecological requirements.

Our results suggest that *A. agrarius*, a species connected to the presence of wood mosaic, uses permanent plantations in highly modified landscapes probably as surrogate of forest habitats. The abundance of this species was negatively related with the amount of arable land so that it becomes absent in very simplified landscapes (*Fischer, Thies & Tscharntke, 2011*). Then, permanent crops could become an important secondary habitat for this rodent in intensive agricultural landscapes.

*A. sylvaticus*, instead, has a broad ecological niche and can exploit many different habitats (*Halle, 1993*; *Michel, Burel & Butet, 2006*), such as fallow lands (*Heroldová et al., 2007*; *Macdonald et al., 2007*) and crops (see for instance *Tattersall et al., 2001*). This species has high dispersal abilities (*Marsh & Harris, 2000*) and is supposed to have higher abilities to navigate in crops (*Sozio, Mortelliti & Boitani, 2013*).

In cultivated landscapes, *A. sylvaticus* moves from hedgerows to crop fields according to seasons (*Ouin et al., 2000*). At the time of the crop, wood mice massively overrun the fields and remain there until the time of ploughing (*Heroldová et al., 2007*); during winter more mice are found in field edges than in the center of fields, and when the crop are cut down they prefer hedgerows, woodland edges (*Macdonald et al., 2000*; *Tattersall et al., 2001*), woods and fallow lands (*Tew & Macdonald, 1993*). This allowed them, to avoid predation, which can bring high rates of mortality (*Tew & Macdonald, 1993*). The wood mouse is therefore well represented in intensified sites because it certainly benefits from landscape heterogeneity as it can find and exploit many resources (*Michel, Burel & Butet, 2006*). As observed by *Bellows, Pagels & Mitchell (2001)* for other rodent species, the lack of relations between the species and microhabitat characteristic is probably because of its ability to exploit a variety of sources.

We did not observe significant relationships between species abundance and percentage of settlements in the surroundings, within the considered buffers. This fact may be

interpreted as the low presence of urban areas, reduced in the study sites to a few scattered agglomerates of buildings (ca 8.5% in a 500 m buffer).

## CONCLUSIONS

Our findings revealed that, in the study areas, landscape composition rather than landscape configuration significantly affect small mammals species in the considered agricultural landscapes, which, along with microhabitat features and species interaction, drives species abundance.

Our results showed that composition of agricultural landscapes is not perceived by all species in the same way. Species respond to each ecological scale in relation to their own ecological requirements, spatial behaviors, mobility, and dependency on habitat (*Kozakiewicz et al., 1999*; *Serafini, Priotto & Gomez, 2019*). The maintenance of wider woodlots (or hedgerows) is, hence, necessary for several aims, as: (i) protecting wildlife species with different ecological needs, (ii) maintaining/enhancing habitat continuity, and (iii) avoiding structure simplification (*Dondina et al., 2016*). Functional connectivity among habitat patches is proven to be one of the most important factors affecting the presence of small mammals in urban areas (*Fitzgibbon, Putland & Goldizen, 2007*) but also matrix composition and management is fundamental since forest species can cross agricultural surroundings (*Mortelliti et al., 2013*; *Paise, Vieira & Prado, 2020*). Our results showed that the presence of anthropogenic habitats in the landscape with plausible landscape connectivity functions (such as permanent plantations) can also affect small mammals.

Studies have been carried on connectivity in the agricultural matrix, but the effect of forest plantations on functional connectivity has received limited attention (*Mortelliti, Westgate & Lindenmayer, 2014*).

Permanent tree crops, especially those aimed at producing woody biomass, have a negative effect on biodiversity when they replace native forest vegetation (*Brockerhoff et al., 2008*; *Greene, Martin & Wigley, 2019*). It is already commonly believed that their effect may have less impact than that of the agricultural matrix, since they are working as a semi-permeable barrier for organisms, representing a compromise between maintaining the economic value of an area and reducing the effect of habitat fragmentation, increasing connectivity (*Norton, 1998*; *Brockerhoff et al., 2008*; *Vanbeveren & Ceulemans, 2019*). Landscape connectivity as well as the habitat function, could greatly increase if management rules are properly applied (*Norton, 1998*; *Lindenmayer, Hobbs & Salt, 2003*; *Zitzmann, Reich & Schaarschmidt, 2021*). This process may guarantee, for example, the permanence of mature plants within plantations (*Hanowski, Niemi & Christian, 1997*) or increase the shrub and herbaceous layers (*Moser et al., 2002*), whose positive effect on small mammals is already known (*Balestrieri et al., 2017*). Even the understory cover and litter can promote biodiversity, if they are not completely eliminated (*Christian et al., 1997*; *Christian et al., 1998*; *Moser et al., 2002*; *Vanbeveren & Ceulemans, 2019*).

Our results showed that several properties of agricultural land mosaics had a strong influence on small mammal abundance, and all of them are within the reach of farmers

management. In this study, we get new insight to support the concept of 'wildlife-friendly farming' (*Fischer et al. 2014*), which emphasizes the ecological interactions of farmed and unfarmed coexisting areas and the improvement of natural and cultivated areas as strategy for an agricultural landscape management aimed to increasing biodiversity. This would contribute to the implementation of policy for the conservation of agricultural landscape at local and regional scales, and to farm landscape design.

## ACKNOWLEDGEMENTS

We gratefully acknowledge Ivano Marcorin, Mattia Spessot, Costanza Uboni for their support during field activities. Thanks to Alessio Mortelliti, Sergio Muratore, Stefania Gentili and Laura Bonesi for helpful comments and precious recommendations.

### Funding

The authors received no funding for this work.

### Competing Interests

The authors declare there are no competing interests.

### Author Contributions

- Luca Dorigo conceived and designed the experiments, performed the experiments, analyzed the data, prepared figures and/or tables, authored or reviewed drafts of the paper, and approved the final draft.
- Francesco Boscutti conceived and designed the experiments, analyzed the data, prepared figures and/or tables, authored or reviewed drafts of the paper, and approved the final draft.
- Maurizia Sigura conceived and designed the analyzed landscape data, or reviewed drafts of the paper, and approved the final draft.

### Animal Ethics

The following information was supplied relating to ethical approvals (i.e., approving body and any reference numbers):

Trapping activities were authorized by the Italian Institute for Environmental Protection and Research (ISPRA) (Aut. PG/Ir Rif. Int. 26358-28967/2013) and by the Friuli Venezia Giulia region (Aut. Prot. SCPA/12.5/17552-2013).

### Data Availability

The raw data are available in the Supplementary File.

### Supplemental Information

Supplemental information for this article can be found online at http://dx.doi.org/10.7717/peerj.12306#supplemental-information.

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
