# Peer review of "Landscape and microhabitat features determine small mammal abundance in forest patches in agricultural landscapes"

_PeerJ, doi:10.7717/peerj.12306_

## Round 0.1 · original submission · Major Revisions

It has been reviewed by experts in the field and we request that you make major revisions before it is processed further.

Reviewer 1 ·

Basic reporting

no comment

Experimental design

The meaning and significance of this study should be further stressed and clarified.

Validity of the findings

no comment

Additional comments

1. Why do you choose to study these two species in this area? Needs more clarification.
2. The meaning and significance of this study should be further stressed and clarified in the Introduction and Discussion part.
3. Conclusions should be more concise, and detailed information may be moved to results and discussion part.
4. In the abstract, context and background should be stated before stating "We studied the populations of two small mammals (Apodemus agrarius, A. sylvaticus) in 19 wood remnants along a gradient of agricultural landscape intensification in Friuli Venezia Giulia (north-eastern Italy)"

Reviewer 2 ·

Basic reporting

See comments below. Generally very good.

Experimental design

Very good.

Validity of the findings

Very good. See minor comments below.

Additional comments

General comments
Globally there are concerns about anthropogenic land-use change effects on biodiversity. In this study, the authors investigated how landscape and microhabitat features determine small mammal abundance (two Apodemus spp.) in forest patches in agricultural landscapes. The work makes an important contribution to our understanding of small mammal species persistence in anthropogenic landscapes.

I suggest replacing ‘due to’ with ‘because of’ throughout.

Specific comments
Title: Fine.

Abstract:
L24 Insert a general sentence.
L25 Maybe say how many woodlots and over what time.

Introduction:
A good synthesis of the relevant literature.
Avoid one-sentence paragraphs, so combine 2nd para with 3rd.
L65 ‘responded’.
L72 Correct author’s name ‘Bricker’.
L82 Replace ‘proved to’ with ‘shown’.
L97 Both references are not in the list.
L108 What were your predictions?


Methods:
Good explanation of methods used.
L116 Insert Latin names.
L214 Put GIS in full at 1st mention.
L214 Is it not QGis?
L218 Correct ‘polygon’.
L234 Change to ‘analyses’.

Results:
Were both species caught in all your sampling sessions?
L291, 294 Avoid one-sentence paragraphs so combine.
L299, 318 Insert space ’Table 2’.
L301 Check Journal format for the presentation of statistics.
L315 ‘3.2’.

Discussion:
Perhaps a bit long.
L518 ‘Conclusions’.
L523, 557 ‘showed’.

References:
I have not checked if the Journal format was followed. Most in-text references are listed.
Please check and follow the Journal format. Some Journal titles are abbreviated, and others not. Some Journal titles do not have each word starting with a capital.

Reviewer 3 ·

Basic reporting

The article is very comprehensive in terms of content and contains all relevant sections. However, I have the following concerns:
1. Some sections (e.g., Materials and methods) contain many grammatical and typographical errors. I have highlighted some of them for the authors' consideration and revision.
2. There are many long sentences (e.g. Lines 551-556) which makes it hard to read.
3. The introduction should be improved to ensure a proper logical flow. For example, there is a sudden change in grammatical tense on Lines 65 and 66. There are also redundant phrases that need to be removed. For example, "on their mosaics" on Line 106.
4. The references are very relevant, but I did not find any recent ones. I noticed that the latest reference is nearly 4 (i.e. 2017) years old. The authors will agree with me that there have been several advances (i.e., important studies) during this period that need to be referenced to improve the quality of the introduction, discussion, and conclusions.
5. I also noted with concern the variable weird names such as "N_spec_arbr", "Dmt_med_fst". The authors should endeavor improve this.
6. There are other editorial revisions that will improve the quality of the paper. For example, a better rendering of the equation on Line 191 with an equation number. I have highlighted most of these editorial errors for authors' review and revision.
7. Table and figure labels are too long in my view. Authors may want to shorten them.

Experimental design

The methodology is sound. no comment

Validity of the findings

No comment

Additional comments

Please see the attached annotated PDF for details.

Annotated reviews are not available for download in order to protect the identity of reviewers who chose to remain anonymous.

---

## Round 0.2 · Minor Revisions

Minor editorial modifications are required. Please see the annotated PDF from Reviewer 3.

Reviewer 1 ·

Basic reporting

Clear and good.

Experimental design

Clear and good.

Validity of the findings

Good.

Additional comments

Revisions have been made according to previous review report, and I think it could be accepted.

Reviewer 2 ·

Basic reporting

See below

Experimental design

See below

Validity of the findings

See below

Additional comments

I thank the authors for their detailed revision. They have addressed all my concerns and I find the manuscript acceptable now.

Reviewer 3 ·

Basic reporting

Minor editorial modifications aimed at improving the readability of the paper have been made for authors' consideration.

Experimental design

no comment

Validity of the findings

no comment

Additional comments

The modified manuscript is attached for authors' consideration

Annotated reviews are not available for download in order to protect the identity of reviewers who chose to remain anonymous.

---

## Round 0.3 · accepted · Accept

I am making a decision on this manuscript as the Ecology Section Editor.

Thanks for your work on this, and for the work done by the reviewers and the Academic Editor up to this point. The manuscript has gone through an adequate review and revision process and the reviewers recommend it as acceptable. I agree with this assessment. Congratulation on your new publication.

Reviewer 3 ·

Basic reporting

no comment

Experimental design

no comment

Validity of the findings

no comment

Additional comments

I thank the authors for their hard work on this paper. Congratulations and best wishes.